# Transcutaneous Vagus Nerve Stimulation Could Improve the Effective Rate on the Quality of Sleep in the Treatment of Primary Insomnia: A Randomized Control Trial

**DOI:** 10.3390/brainsci12101296

**Published:** 2022-09-26

**Authors:** Yating Wu, Lu Song, Xian Wang, Ning Li, Shuqin Zhan, Peijing Rong, Yuping Wang, Aihua Liu

**Affiliations:** 1Neurology Department, Xuanwu Hospital, Capital Medical University, Beijing 100054, China; 2Neurology Department, Beijing Tsinghua Changgung Hospital, Beijing 102218, China; 3Rehabilitation Medicine Center, Fuxing Hospital, Capital Medical University, Beijing 100038, China; 4Institute of Acupuncture and Moxibustion, China Academy of Chinese Medical Sciences, Beijing 100700, China

**Keywords:** transcutaneous vagus nerve stimulation, primary insomnia, Pittsburgh Sleep Index, anxiety, depression

## Abstract

(1) Background: The purpose of this study was to investigate the efficacy and safety of transcutaneous vagus nerve stimulation (t-VNS) in the treatment of primary insomnia. (2) Methods: This is a single center, randomized, double-blind study. A total of 30 patients diagnosed with primary insomnia were randomly divided into two groups to receive 20 Hz t-VNS in either the auricular concha area (treatment group) or periauricular area (control group), twice a day for 20 min during a one-month study period. The effective rate of treatment, defined as a ≥50% reduction of the Pittsburgh Sleep Index Scale (PSQI) after treatment, was compared between the two groups as the primary outcome. Response rate (defined as ≥10% change in the PSQI score), and changes in the Hamilton Anxiety Scale (HAMA) and Hamilton Depression Scale (HAMD) scores were also assessed. (3) Results: After one month of treatment, the PSQI score of the treatment group decreased significantly (*p* = 0.001). The effective rate of the treatment group (73% vs. 27%, *p* = 0.027) was significantly higher than that of the control group. No statistical differences in changes of HAMA and HAMD scores were detected between the two groups. There were no complications in all patients. (4) Conclusion: T-VNS appeared to be a safe and effective treatment for primary insomnia.

## 1. Introduction

Insomnia is the most common sleep disorder, while primary insomnia refers to insomnia caused by no clear cause. The prevalence of insomnia has surged with the changes in the pace of modern life. Recent studies have suggested that the estimated prevalence of insomnia is 22–64.2% in the general population [1,2,3], accompanied by a slight rise in consideration of gender and age difference [3,4]. Short-term insomnia can lead to fatigue, daytime sleepiness, and work inefficiency [5]. Meanwhile, long-term insomnia can lead to emotional instability and increased risk of chronic diseases such as cardiac disease, cancer, and can even result in increased mortality [6,7,8]. At present, treatment measures to counter against insomnia comprise cognitive behavioral and medical therapies, including benzodiazepines and non-benzodiazepines, together with alternative therapy such as acupuncture [9,10,11]. However, clinical use of medical drugs for sleeping disorders is often practiced cautiously with apparent concerns regarding the incidence of adverse events such as dependency and drowsiness [12,13,14].

At present, based on anatomy, theoretically, it has been suggested that the stimuli towards the parasympathetic system activates the solitary tract nucleus and projects fibers to the central sleep structures such as the parabrachial nucleus, ventrolateral hypothalamic preoptic area (VLPO), anterior median nucleus (MnPo), locus ceruleus, raphe nucleus, reticular structure, thalamus, etc. [15]. Among them, activation of the reticular structure could lead to the genesis of slow-wave sleep. Vagus nerve stimulation can also participate in the regulation of sleep by changing the concentration of neurotransmitters such as γ-aminobutyric acid (GABA), norepinephrine (NE), and 5-hydroxytryptamine (5-HT). Vagus nerve electrical stimulation (VNS) has been used in the treatment of refractory epilepsy, depression, and insomnia [16,17,18].

Devices for vagus nerve stimulation can be categorized into different types according to their installation approach. Implantable devices were earlier products than transcutaneous devices; however, major concerns regarding laryngeal dysfunction, cardiac side effects, and sleep apnea have limited the clinical use of implantable devices [19,20,21]. Transcutaneous vagus nerve stimulation (t-VNS) devices were first conceptualized by Ventureyra, and they have been proven to demonstrate equivalent efficacy as implantable devices, accompanied by other advantages such as low cost, non-invasiveness, less incidence of adverse effects, portability, and no requirement of battery change [22,23,24]. Transcutaneous devices can be divided into auricular and cervical approach (taVNS and tcVNS, respectively) devices [22]. The anatomical basis for trans-auricular devices is that the auricular branch, as the only surficial branch of the vagus nerve, originates from the upper cervical ganglion of the vagus nerve and distributes its surrounding branches in the ear concha cavity and the external auditory canal [25]. Transcutaneous devices have been successfully applied in resistant epilepsy, major depression, post-stroke rehabilitation, and the general geriatric population [26,27,28]. All the above studies have reported improved quality of life and sleep observed in participants. However, only a few studies have been carried out to evaluate the long-term efficacy and safety of taVNS in primary insomnia [18], and therefore, there is a need for more research. The purpose of this study is to explore the efficacy and safety of transcutaneous vagus nerve stimulation in the treatment of primary insomnia.

## 2. Methods and Materials

### 2.1. Study Design

This was a single-center, prospective, double-blind, randomized, controlled trial. From January 2016 to April 2017, patients diagnosed with primary insomnia at the Sleep Clinic of the Department of Neurology of Xuanwu Hospital, Beijing, China were enrolled and randomly assigned into two parallel groups. The patients in the treatment group received 20 Hz t-VNS in the auricular concha area, while the patients in the control group received the same stimulation in the periauricular area; both groups received treatment twice a day for 20 min during an 1 month study period. The primary outcome assessed was the effective rate of treatment, defined as ≥50% reduction in the Pittsburgh Sleep Index Scale (PSQI) score after treatment. The study was approved by the local ethics committee and all participants signed informed consent. A portion of the data of this study has been published in a previous study and was used here with the authors’ permission [18]. This trial is registered at the Chinese Clinical Trial Registry with the registration number ChiCTR-TRC-13003519.

### 2.2. Patients

During the study period, consecutive patients seeking medical consultation at the sleep clinic due to insomnia symptoms were screened by experienced doctors who had majored in sleep medicine for eligibility. The inclusion criteria included: (1) Met the diagnostic criteria of primary insomnia according to the third edition of the International Classification of Sleep Disorders (ICSD-3); (2) Pittsburgh Sleep Index Scale (PSQI) scores >7; (3) Hamilton Depression Scale (HAMD) scores <20, and Hamilton Anxiety Scale (HAMA) scores <14; (4) no medicine that affected sleep was taken 1 month before entering the group, or the type and dose of drugs were not adjusted 1 month before entering the group; (5) age between 18 and 70 years old; (6) understood the contents of the scale, agreed to cooperate with treatment and follow-up, and voluntarily participated in this study. The exclusion criteria were as follows: (1) Pregnant, breastfeeding and menopausal women; (2) complicated with serious diseases including heart, liver, kidney and blood system, and somatic diseases including hyperthyroidism and chronic pain; (3) secondary insomnia, i.e., concomitant symptoms of somatic or other mental diseases and sleep disorders caused by poor sleep hygiene, substance abuse, or other causes; (4) those who were receiving relevant treatment that was considered to affect the study outcome.

### 2.3. Interventions

The technical route of this study is shown in Figure 1. After the baseline information was recorded, patients underwent an initial evaluation with the neuropsychological scale, and then were treated with t-VNS. This study was both randomized and double-blinded. Patients were randomly divided into treatment and control groups, and only after the completion of the experiment, the patients and the experimenter who took part in the study could be unblinded.

Regarding stimulation location, in the treatment group, the bilateral auricular concha area (including auricular boat and cavity) was selected, and the stimulation area was about 3 cm^2^. In the control group, the bilateral scapha of the auricle was selected, which is not innervated by the auricular branch of the vagus nerve but rather by the lesser occipital nerve (Figure 2). The stimulation instrument applied transcutaneous vagus nerve stimulation (TENS-sm, Suzhou Medical Equipment Co., Ltd., Suzhou, China) by connecting an electrode clamp with TENS through a metal wire. The parameter settings were: pulse frequency of 20 Hz, pulse width of 0.2 ms, current of 1mA, and stimulation was intensity adjusted according to the maximum intensity that patients could tolerate. After the training of professionals, patients underwent stimulation as a routine treatment every day, 20 min each time, twice a day. The total treatment period was 1 month.

### 2.4. Outcomes

The primary outcome was the effective rate of treatment. Effectiveness was defined as a final reduction in the PSQI score of ≥50% at the end of study as compared with the baseline PSQI score. In addition, response rate was defined as the percentage of patients with a reduction in the PSQI score of ≥10%. Other secondary outcomes included the changes in the anxiety and depression symptom scores. During the treatment, in a sleeping diary, adverse events were recorded every day; the PSQI scores were evaluated ach week untill the end of treatment. PSQI, HAMA, and HAMD scores were re-evaluated at the end of the study.

### 2.5. Estimation of Sample Size

According to the results of a small sample preliminary experiment, the effective rate of the treatment group was 60% (5 cases, 3 of which were effective), and the effective rate of the control group was 20% (5 cases, 1 of which was effective). At the settings with a significant level of 0.05, and expected test efficiency of 0.90, substituting the sample calculation formula *n* = (+)2 × (1 + 1/k) × *P* (1 − *p*)/(*P*1 − *P*2) 2, the sample sizes of the treatment and control groups were 26 cases, respectively. Taking into consideration a 20% dropout, 62 cases were needed. An interim analysis was planned for 32 patients. As patient recruitment was slower than expected, the sponsor decided to discontinue the study at the time of the interim analysis, and therefore, did not continue to recruit patients.

### 2.6. Statistical Analysis

The two groups were compared with respect to the following: baseline data (age, gender, course of disease, and family history), response rate, effective rate, PSQI scores at each week, and HAMA/HAMD scores before and after treatment. Categorical variables were represented by counts (percentages). Continuous variables that did not conform to the normal distribution were represented by the median (IQR). The continuous variable of normal distribution is expressed using the mean ± SD. A chi-square test or Fisher exact test was used to compare the differences between categorical variables. The Mann–Whitney U test was used to compare the differences between non-normal distribution continuous variables, while an independent double sample t-test was used to compare the differences between normal distribution continuous variables. When the measured data of the same group before and after treatment were in accordance with a normal distribution, a paired t-test was used; a nonparametric test was used when it did not conform to a normal distribution. All statistical assessments were two-sided tests, and a *p*-value < 0.05 was considered to be significant. Version 25.0 of the SPSS software was used (SPSS Inc., Chicago, IL, USA).

## 3. Result

A total of 32 patients with primary insomnia were enrolled in this study, including 16 patients in the experimental group and 16 patients in the control group. A total of 30 patients completed the four-week t-VNS treatment and two patients dropped out of the study. (Figure 3). The reasons for dropping out included poor therapeutic effect (one case) and insomnia drug intake (one case).

The 30 patients in the final analysis included six males and 24 females. The average age was 45.4 ± 12.2 years old, and the average course of disease was 54.7 ± 66.4 months. The cause of insomnia was unknown in 26 patients, four cases were caused by psychological and mental reasons, and five patients had a family history of insomnia. There was no significant difference in the baseline characteristics between the treatment control groups (shown in Table 1).

Among the 30 patients who completed the study, a total of 14 (93.3%) patients in the treatment group had decreased PSQI scores after t-VNS treatment, and one patient had no improvement. Among the 15 patients in the control group, 11 patients (78.6%) had decreased PSQI scores. By comparing the PQSI scores before and after treatment, it was found that the scores in the treatment group decreased significantly from the first week after treatment, while the control group decreased significantly from the third week after treatment. After treatment, the PSQI scores of both the treatment group and the control group decreased significantly(Table 2).

In this study, the response rate and effective rate of treatment were calculated to evaluate the efficacy of t-VNS in the treatment of primary insomnia. When the median PSQI scores between the treatment and control groups were compared at each assessment, there were no significant differences detected (shown in Table 3).

However, the response rates in the treatment group were higher than those of the control group (33% vs. 60%, 67% vs. 87%, 73% vs. 93%, and 73% vs. 93%) at 1 week, 2 weeks, 3 weeks, and 4 weeks after treatment, respectively. In addition, the effective rates in the treatment group were also higher than those of the control group (0 vs. 13%, 7% vs. 20%, 13% vs. 33%, and 27% vs. 73%) at 1 week, 2 weeks, 3 weeks, and 4 weeks after treatment, respectively. However, only the effective rates of the treatment group at 4 weeks were significantly higher than those of the control group (shown in Table 4).

The anxiety and depression scale scores of patients in the treatment group and the control group were decreased one month later after treatment. However, there was no significant difference between these two groups (shown in Table 5). The HAMA and HAMD scores of patients in the two group decreased significantly after treatment as compared with the baseline scores. In addition, after the t-VNS treatment, none of the 30 patients in the study had adverse reactions.

## 4. Discussion

Sleep is influenced by emotions, and also by both neural and humoral regulation, in which the vagus nerve plays an important role in the regulation of sleep and mood owing to its extensive distribution and functions [29]. At present, VNS is an effective adjuvant treatment for epilepsy, depression, and insomnia. Some patients receiving VNS have experienced improved quality of sleep and life, reduced daytime sleepiness, and improved mood. Moreover, these changes were independent of the anti-epilepsy or anti-depression effects of VNS, suggesting that VNS could regulate sleep. Moreover, a previous study that focused on the application of taVNS modalities for insomnia suggested that improved sleep quality could be observed in patients [30]. Taken together, therapies including taVNS could be more beneficial to patients.

In this study, there were no significant differences between the ta-VNS group and tn-VNS in changes of primary and secondary outcomes, but the PSQI scores of the treatment group decreased significantly after ta-VNS treatment, which was consistent with Rong’s study [18]. Moreover, we included different outcomes, including effective rate and response rate. In addition, our findings suggested the effective rates of the treatment group were significantly higher than those of the control group, which provides new evidence for the evaluation of the effect of ta-VNS in the treatment of insomnia. Other studies have revealed that vagus nerve stimulation could interfere with the synchronization and desynchronization of the cortex, increase the number of periods in the REM phase and PGO waves, and prolong the REM phase. Similar to the findings of sleep monitoring in cats, it was found that VNS could also increase the number of spindles and δ waves, and prolong the total night sleep time, especially the second stage sleep time. After a 6 month follow-up of patients treated with VNS, Galli et al. found that low intensity (<1.75 mA) VNS could prolong the average sleep latency, shorten N1 phase sleep, and increase the proportion of slow wave sleep [31]. The results of another study also demonstrated a sharp decline in PSQI scores after two weeks of treatment [30]. In our follow-up of patients that lasted for four weeks, we also observed a similar decline in PSQI scores. Moreover, a continuous trend of declining PSQI scores could be seen in the following weeks of experimentation.

However, in this study, the sleep quality of patients in the control group was also improved. The reasons could be the following: It is known that the formation and maintenance of insomnia is related to cognitive and psychological factors, including worrying about the influence of insomnia and excessive thinking at night [32]. Patients in the control group displayed psychological cues by using the t-VNS instrument, and therefore, achieved the effect of improving insomnia. In addition, the stimulation area was located in the periauricular area and close to the auricular concha area where the peripheral branches of the vagus nerve were distributed in the control group, and the stimulation of the periauricular area could also affect the afferent of the vagus nerve. In addition, there were two electrode clips in the ear clip of the taVNS instrument, in which one was located in the periauricular area and the other was located in the auricular concha area for the control group, although the electrode clip in the auricular concha area had no release of electrical current, its pressure stimulation could also affect the vagus nerve afferent. Thus, the improved outcome for patients in the control group could be contributed by psychological cues resembling placebo effects.

Insomnia is often accompanied by anxiety and depression; anxiety and depression cn often aggravate insomnia. It has been confirmed that emotional improvement could improve sleep efficiency and sleep satisfaction. Klinkenberg et al. found that the results of follow-up of epileptic patients for 6 months who received VNS, the mood and quality of life of the patients were significantly improved [33]. Rong et al. confirmed that VNS could effectively reduce the degree of depression in patients, and the effect was positively correlated with the time of use [34]. In this study, the HAMA and HAMD scores of patients in the two groups decreased significantly as compared with the baseline period, which was consistent with previous studies. However, there was no significant difference between the experimental group and the control group, which indicated that psychological cues could significantly affect the mood of patients in the control group.

The PSQI used in this study was designed in 1989, in which higher scores suggest poorer sleep; a relevant meta-analysis of PSQI scores has demonstrated that it had strong validity and reliability for the evaluation of insomnia severity [35]. The HAMA scoring system has been widely adopted for the assessment of anxiety [36]. The HAMD scoring system has been used for the evaluation of depression, and has also shown generally sound discrimination power in the stratification of patients according to the severity of disease [37,38].

This study also had some limitations. First, the sample size in this study was relatively small. This might be due to the strict inclusion criteria of PSQI degree and course of disease. In addition, the follow-up time was short, considering the control group was set up in this study, further follow-up observation was needed in the later stage. Furthermore, a lack of objective evidence such as results of imaging and electroencephalogram may also interfere with the interpretation of the actual conditions in our participants. Taken together, these questions are waiting for further studies in the future.

## 5. Conclusions

In conclusion, the results of this study show that auricular t-VNS at 20 Hz for 1 month is effective for primary insomnia patients; the patients obtained an absolute reduction in PQSI scores. The t-VNS treatment also significantly relieved the level of anxiety and depression assessed by the HAMD and HAMA scores, with good safety and high compliance with daily stimulation.

## Figures and Tables

**Figure 1 brainsci-12-01296-f001:**
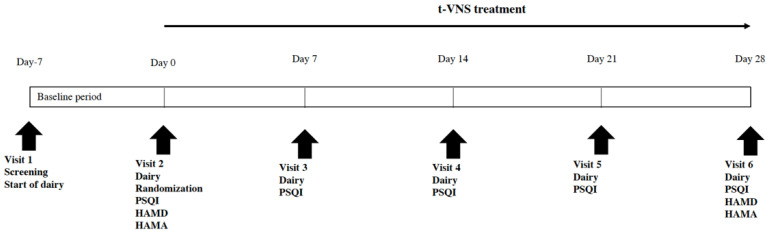
The timeline diagram of the study process, which demonstrates the whole process of study including the follow-up.

**Figure 2 brainsci-12-01296-f002:**
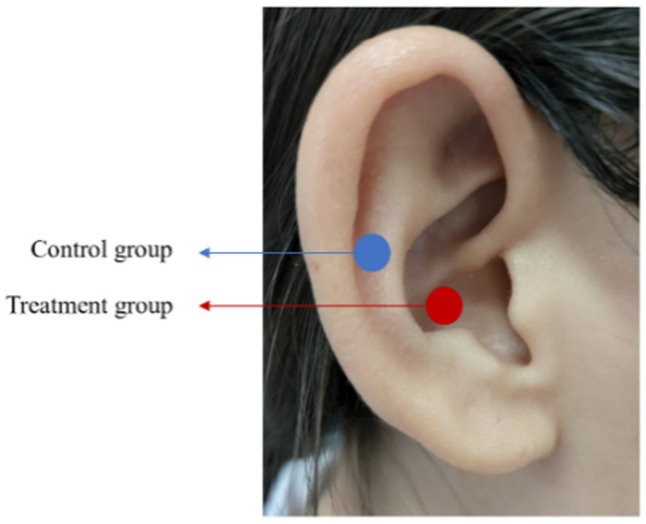
Positioning of the electrode of the t-VNS device for stimulation in the treatment and control groups.

**Figure 3 brainsci-12-01296-f003:**
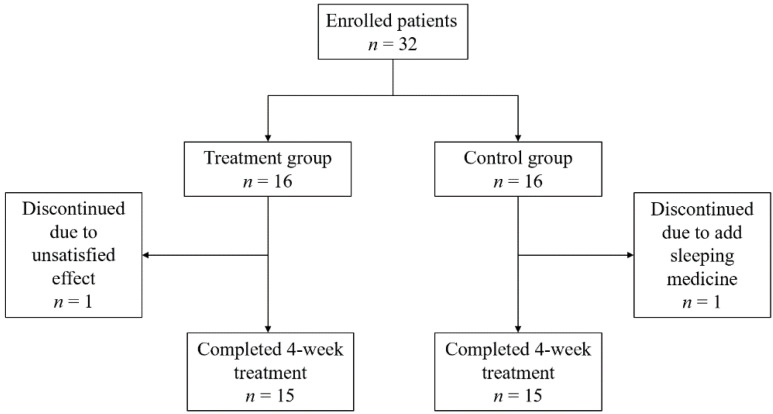
A flowchart of the study. The number of enrolled patients was 32 in total, and equal numbers of patients were randomly grouped into treatment and control groups. One patient in the treatment group discontinued the study due to unsatisfactory effects associated with the therapy, and another patient dropped out from our study because of adding sleeping medicine.

**Table 1 brainsci-12-01296-t001:** Comparisons between treatment and control groups, at baseline.

	Treatment Group (*n* = 15)	Control Group (*n* = 15)	*p*
Gender			
Female	13	11	0.651
Age (mean ± SD)	45.9 ± 15.4	47.1 ± 13.2	0.687
Course of disease (months)	52.5 ± 38.6	59.1 ± 43.5	0.253
Family history	1	3	0.598
Baseline PSQI (median, IQR)	14 (9–18)	12 (8–16)	0.791
Baseline HAMD (mean ± SD)	13.4 ± 3.5	15.6 ± 6.2	0.458
Baseline HAMA (mean ± SD)	15.9 ± 4.7	12.1 ± 5.3	0.443

**Table 2 brainsci-12-01296-t002:** Results of weekly PSQI scores for the treatment group and the control group.

	Baseline	Week 1	Week 2	Week 3	Week 4
Treatment group					
PSQI (median, IQR)	14 (9–18)	11 (7–13)	8 (5–11)	8 (4–10.5)	6 (3–9.5)
Z	1 (referent)	−2.045	−3.303	−3.416	−3.353
P		0.041	0.001	0.001	0.001
Control group					
PSQI (median, IQR)	12 (8–16)	11 (6–13)	9 (5–12)	9 (5–12.5)	7 (4–11)
Z	1 (referent)	−0.709	−1.866	−2.166	−2.269
P		0.478	0.062	0.030	0.023

**Table 3 brainsci-12-01296-t003:** Comparison of the PSQI scores between the treatment and control groups.

	Treatment Group (*n* = 15)	Control Group (*n* = 15)	*p*
PSQI (median)			
Baseline	14 (9–18)	12 (8–16)	0.791
Week 1	11 (7–13)	11 (6–13)	0.982
Week 2	8 (5–11)	9 (5–12)	0.652
Week 3	8 (4–10.5)	9 (5–12.5)	0.534
Week 4	6 (3–9.5)	7 (4–11)	0.422

**Table 4 brainsci-12-01296-t004:** Comparison of the effective rates and response rates between the treatment and control groups after the treatment.

	Week 1	Week 2	Week 3	Week 4
Response rate				
Treatment group	9 (9/15.60%)	13 (13/15.87%)	14 (14/15.93%)	14 (14/15.93%)
Control group	5 (5/15.33%)	10 (10/15.67%)	11 (11/15.73%)	11 (11/15.73%)
*p* value	0.143	0.390	0.330	0.330
Effective rate				
Treatment group	2 (2/15.13%)	3 (3/15.20%)	5 (5/15.33%)	11 (11/15.73%)
Control group	0%	1 (1/15.7%)	2 (2/15.13%)	4 (4/15.27%)
*p* value	NA	0.589	0.390	0.027

**Table 5 brainsci-12-01296-t005:** Comparison of the HAMD and HAMA scores between the treatment and control groups.

	Treatment Group (*n* = 15)	Control Group (*n* = 15)	*p*
HAMD			
Final (mean ± SD)	8.2 ± 4.4	7.5 ± 3.0	
Change	−5.3	−6.5	0.324
HAMA			
Final (mean ± SD)	6.5 ± 5.0	6.1 ± 4.0	
Change	−3.1	−3.2	0.809

## Data Availability

The datasets generated during and/or analyzed during the current study are available from the corresponding author on reasonable request.

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
