# Peer review of "Transcutaneous Vagus Nerve Stimulation Could Improve the Effective Rate on the Quality of Sleep in the Treatment of Primary Insomnia: A Randomized Control Trial"

_brainsci, 2022, doi:10.3390/brainsci12101296_

Round 1
Reviewer 1 Report
This is a randomized, parallel, controlled, double-blinded study. Thirty insomnia patients received 20-min 20Hz t-VNS stimulations twice a day for 4 weeks. The stimulations were delivered over auricular concha area and periauricular area in the treatment and control groups, respectively. PSQI, HAMA and HAMD were assessed. The effective rate (>=50% reduction of PSQI by end-of-treatment) and response rate (>=10% reduction of PSQI) were calculated. Significant reductions of PSQI were found in the treatment group since week 1 and in the control group since week 3. There was no significant difference in median PSQI scores between the two groups throughout the treatment. In terms of both response rate and effective rate, they were higher in the treatment group than in the control group at week 1 – 4.
Evaluating the efficacy of t-VNS for treating insomnia is of clinical importance. The current and previous research from this group of authors suggests t-VNS appears to be a safe and effective treatment for primary insomnia. In the current study, the lack of difference of PSQI changes between treatment and control groups concerns me, although there were significant differences between the groups in effective rate and response rate. Also, the authors claimed this is a single-site study collecting data from January 2016 through April 2017. Some of the authors also published a multi-center study with similar study design and purposes. It’s not clear what’s the association between these two studies. I think there are some other issues need to be addressed as well. Unfortunately, I cannot recommend the publication of this manuscript in its current form. My comments are as follows:
1, The authors published several studies using t-VNS for treating primary insomnia. For example, Peijin Rong reported a multi-center clinical trial on this topic in 2020 using a similar study design (Jiao et al., 2020; title: Effect of Transcutaneous Vagus Nerve Stimulation at Auricular Concha for Insomnia: A Randomized Clinical Trial). It seems that the ‘single site’ in the current study might be one of the sites in the multi-site study mentioned above. Could the authors clarify if this is the case and if any data reported here are part of the data reported in previous publications?
2, following on the above comment, the findings from the current study did not seem introducing additional knowledge compared to the above-mentioned multi-center study with a larger sample size published by some of the authors. If the authors think otherwise, please provide such information in the Introduction.
Also, it is not clear why many of the papers on the same topic were not mentioned or cited in this manuscript and some of the papers were from this group of authors. For example, Zhao et al., 2020 (titled: The Instant Spontaneous Neuronal Activity Modulation of Transcutaneous Auricular Vagus Nerve Stimulation on Patients With Primary Insomnia); Wu et al., 2021 (titled: Brain Functional Mechanisms Determining the Efficacy of Transcutaneous Auricular Vagus Nerve Stimulation in Primary Insomnia).
3, Although the treatment group (ta-VNS) and control group did not show significant difference in the change of PSQI scores, ta-VNS did cause more reduction of PSQI than its control group in previous studies (e.g., Jiao et al., 2020). Could the authors discuss the potential reasons for the discrepancy? Also, I wonder if the authors could compare the two groups in terms of the sub-components of PSQI questionnaire. This may provide additional information, such as whether ta-VNS may be more effective in certain sub-components of sleep than the controls.
4, it was mentioned that participants received two treatments per day. What’s the interval between these two treatments? Is the interval consistent throughout the treatment?
5, Independent-sample and paired-sample t tests were used for most of the comparisons. It may be more appropriate to conduct the linear mixed effect model to evaluate the group by time interaction first. Then, use t tests for post-hoc analysis.
6, In the Discussion section, it was mentioned that ‘In this study, the HAMA and HAMD scores of patients 256 in the treatment group decreased significantly after treatment, and the difference was statistically significant compared with the baseline period …’. Unless I missed it, this result seemed not mentioned in the Results section. Please either remove this statement in the Discussion section if this is not true or add this to the Results section.
7, on page 9, line 273, it was stated that ‘lack of subjective evidences such as results of imaging and electroencephalogram’. Do you mean ‘objective evidence’ rather than ‘subjective evidences’?
8, is this a clinical trial? If so, is it registered anywhere (e.g., Chinese Clinical Trial Registry)?
Reviewer 2 Report
The manuscript by Wu et al. investigated the efficacy of transcutaneous vagus nerve stimulation (t-VNS) in on primary insomnia, and concluded that this treatment is effective based on improved in PSQI only in treatment group. Below are major concerns over the trail design, conclusion and authors’ writing.
1. Authors need to improve their writing. There are many errors in scientific terminologies throughout the manuscript that must be fixed. For example:
1) in the title “Transcutaneous Vague Nerve Stimulation” should have been Vagus nerve.
2) in the introduction authors wrote “nervus sympatheticus”, is it sympathetic nervous system?
3) in the introduction “5-hydroxytryptamine (5-HE)”, the abbreviation should have been 5-HT.
Besides, many sentences in the manuscript are phrased awkwardly, please proof read and rephrase some sentences. For example, in the introduction, authors wrote “primary insomnia refers to insomnia caused by no clear cause”.
2. The authors claim that they put electrode in auricular concha area for treatment group and periauricular area for control group. They show a picture of device position in Figure 2 for treatment group. This figure is taken from the internet without citation, https://me-pedia.org/wiki/File:TranscutaneousVagusNerveStimulation.png
This is unacceptable.
3. Where is the control electrode placed? How do you confirm that the position between treatment and control group are different? The fact that control group also experienced significant PSQI improvement post treatment suggest that the electrode may have been misplaced. The authors need to provide additional evidence showing how electrodes are placed differently between groups.
3. Regarding the conclusion, the authors claim that t-VNS is effective in treatment primary insomnia. However, as Table 3 showed that PSQI scores between two groups are very similar at each given timepoints, it is hard to draw such conclusion. The authors may want to consider including more patients or using more than PSQI as measurement for treatment effectiveness.
Reviewer 3 Report
The current manuscript is a RCT examines efficacy and safety of transcutaneous vagus nerve stimulation (t-VNS) in the treatment of primary insomnia. Please find the following comments regarding the manuscript:
1- Overall, the writing style in the manuscript needs reconsideration.
2- Introduction:
A- L 37-42: " At present, treatment measures used to counter against insomnia comprises cognitive behavioral therapies and medical therapy, including ben zodiazepines and non-benzodiazepines, together with alternative therapy such as acu- puncture[9-11]. However, clinical usage of medical drugs for sleeping disorders is often 40 cautious with apparent concern on the incidence of adverse events such as dependency 41 and drowsiness[12-14].'' This does not seem sound to mention the current interventions for the problem, and main limitations.
B- L 43-51: '' At present, anatomy theoretically suggests that the stimuli towards nervus sympathet icus could activate the nucleus of solitary tract and project fibers to the central sleep structures such as parabrachial nucleus, ventrolateral hypothalamic preoptic area (VLPO), anterior median nucleus (MnPo), locus ceruleus, raphe nucleus, reticular structure, thalamus ,etc[15]. Among them, activation of reticular structure could lead to the genesis of slow wave sleep. Vagus nerve stimulation can also participate in the regulation of sleep by changing the concentration of neurotransmitters such as γ-aminobutyric acid (GABA), norepinephrine (NE) and 5-hydroxytryptamine (5-HE). Vagus nerve electrical stimulation (VNS) has been used in the treatment of refractory epilepsy, depression and other disease[16-18].'' This is not sufficient to justify the relations between vagus nerve stimulation and sleep disorders C- Please provide examples about vagus nerve stimulation and sleep disorders rather than stroke, and epilepsy.
3- Methods:
A- Lines 74-78 are one sentence. it is very long.
B- The reporting of methods does not seem true. Please report the study according to CONSORT statement.
C- Is this clinical trial registered?
Round 2
Reviewer 1 Report
This group of authors has done outstanding research on this topic. However, in the current manuscript, there are only marginal and incremental improvements. Also, if the data (or part of the data) have been published elsewhere, it is critical to mention it in the main text.
Reviewer 2 Report
The revised manuscript addressed all of my concerns except for the last one, which I expand below.
The authors used three different indexes to examine the effect of the treatment on sleep improvement between experimental and control groups, these three indexes are: PSQI, response rate, and effective rate. The smallest p values (week 4) between groups of these three indexes are 0.422, 0.330 and 0.027, respectively. So 2 out of 3 indexes did not show significant difference (p value < 0.05, which is the lowest standard), and the two 'negative' indexes include the one—PSQI—that has strong validity and reliability in the field. Therefore, I find the authors reaching a strong conclusion as stated in the current title is far-fetched. However, if the authors are willing to weaken their title and add in the key word 'effective rate', something like 'Transcutaneous Vague Nerve Stimulation could improve the effective rate on the quality of sleep in the Treatment of Primary Insomnia, a ran-3 domized control trial' , I would then find the statement well supported and the manuscript good for publication.
Reviewer 3 Report
The authors did not address the previous comments in the first report. The manuscript is still suffer from major misleading reporting.
